# Desert Ant (*Melophorus bagoti*) Dumpers Learn from Experience to Improve Waste Disposal and Show Spatial Fidelity

**DOI:** 10.3390/insects15100814

**Published:** 2024-10-16

**Authors:** Sudhakar Deeti, Ken Cheng

**Affiliations:** School of Natural Sciences, Macquarie University, Sydney, NSW 2109, Australia; ken.cheng@mq.edu.au

**Keywords:** insect behaviour, ant colony, waste management, naive ants, experienced dumpers, sector fidelity

## Abstract

Eusocial insects maintain hygiene in their nests with cleaning behaviours. One such behaviour is the disposal of refuse material, food waste, dead nestmate bodies, and other waste, outside the nest. This study on the waste disposal behaviour of Central Australian red honey-pot ants, *Melophorus bagoti,* provides valuable insights into the significance of learning in waste disposal. Naive ants improved over five consecutive runs in waste dumping, running straighter paths with less scanning. The observed differences between experienced and naive ants in waste dumping imply that learning contributes to more streamlined and less time-consuming waste management. In addition, dumping ants established sector fidelity in the consecutive runs, heading mostly in the same general direction on each dumping trip. This behaviour reduces the amount of navigational expertise that the dumper has to acquire. Learning in individuals thus helps an ant colony to cope with not only navigational challenges in foraging, which have been well studied in this and other desert ant species, but also with other demands of living in a harsh desert habitat.

## 1. Introduction

Waste poses a significant threat to the health of social groups due to the potential transmission of diseases. In response to this risk, various animals, including those in eusocial societies with large colonies, have evolved behavioural strategies to manage waste. Eusocial insects, such as ants and eusocial bees, generate substantial waste, including excreta, food remnants, cocoon casings, and deceased members (ants: [1,2,3,4,5]; stingless bees: [6]; other subsocial insects: [7]). The accumulation of such waste in proximity to colony members increases the likelihood of infections and the spread of diseases caused by fungi, bacteria, or parasites [8]. While waste management in insects has been much studied, little focus has been placed on learning processes in waste-management behaviour. Capitalizing on the recently documented waste-dumping behaviour of Central Australian red honey-pot ants, *Melophorus bagoti* [9], we examined differences between experienced and naive ants in waste dumping and followed naive dumpers over multiple runs to gain insights into the role of learning in waste disposal.

Before foraging, inexperienced naive red honey ants walk around the nest to acquire visual information about the surrounding terrestrial panorama through a series of learning walks [10]. This knowledge is then employed for navigation when the ant commences foraging. Foraging ants also use another well-studied navigational strategy of path integration (PI), through which they continuously keep track of a vector pointing to their starting point [11]. When travelling, the animals combine their compass information with estimates of speed or distance constantly to update a home vector, which they can use at any time to guide them directly back to their starting point of journey, typically the nest [12,13,14,15]. When ants are uncertain about the nest position, or in the absence of PI and learned visual cues, foragers employ systematic search as a back-up mechanism, in which they perform looping search walks until they find information, whether local, panoramic, or celestial, which will aid navigation [16,17,18,19]. While we know much about the naive learning walks of foragers, we do not know how naive dumpers overcome the navigational challenges in taking waste outside and returning home again in a visually cluttered environment.

One behaviour thought to facilitate view learning on learning walks is stopping to look. Naive ants on learning walks frequently engage in body rotations with saccades to fixate in different directions, among other things to enable them to learn and remember the nest panorama. Previous studies on desert ants, such as *Cataglyphis noda*, *Cataglyphis aenescens*, and *Melophorus bagoti*, have identified two types of body rotations: pirouettes and voltes [10,19]. During pirouettes, ants turn their bodies completely (±360°) or partially (±180°) to face different directions, intermittently stopping and fixating their gaze for at least 100 ms. This behaviour likely facilitates the learning of views of the panoramas in different directions around the nest entrance. In voltes, ants walk a small circle in a rotational movement without stopping, potentially allowing the calibration of celestial cues for navigational purposes. However, the precise role of these turns in navigation remains unclear.

Red honey ants dump waste regularly [9], but how naive ants, unfamiliar with their surroundings, learn enough navigational information to successfully carry out dumping is largely unknown. Red honey ants disposed of different experimentally provided materials at different distances. Except for ant larva shells, the items of animal origin were dumped at average distances exceeding 5 m. Conversely, items with little or no animal content were dropped at average distances less than 1.2 m. Here, we set out to examine learning processes in waste dumping for the first time in red honey ants by characterizing *M. bagoti* workers’ paths in their dumping behaviour.

The red honey-pot ant *Melophorus bagoti* is the most thermophilic ant on the Australian continent [20]. The ants perform outdoor activities during the daytime on hot summer days, the activity ceasing during the twilight evening, entire night and early morning. Each evening a single worker closes the nest entrance from the outside and remains stationed there until the next day when the ants inside open the entrance (personal observations). Foragers primarily scavenge deceased arthropods but also collect sugary plant exudates and seeds [21,22]. Foraging ants work individually for short durations, covering distances up to 50 m from the nest, relying on path integration and terrestrial visual landmarks without utilizing any chemical trails [23]. Besides foraging, some workers dedicate themselves to excavating the nest throughout the day. Nest excavators remove sand from the nest, depositing it within a 10 cm radius outside [24,25]. In waste management, these red honey ants employ two distinct strategies: transporting waste materials away from the nest during the active above-ground summer months and depositing waste in subterranean latrine channels/tunnels, approximately 10 to 12 cm below ground level, a behaviour likely persistent throughout the year (personal observations) [9].

The limited extant data on dumping behaviour in ants leave key open questions to be examined. A chief open question is the visual knowledge of the naive dumpers. Can they home from a range of directions near the nest? To test such knowledge of the visual surroundings, the standard and accepted method is to give displacement tests to returning ants that have nearly reached to their nest. On an outbound job, whether foraging, excavating, or dumping, desert ants can call on multiple strategies in their navigational toolkit. They can use path integration (PI) to continuously track the distance and compass direction from their starting point (the nest) of all path segments during the trip ([26,27,28]; in *M. bagoti*: [23,29]). PI enables an animal to orient and face in the direction of the starting point without any knowledge of the nest’s visual surroundings. If an ant has just about returned to its nest, however, its PI vector is near 0 and is not informative as to a direction of travel. Displacement tests force an ant to use another strategy in the toolkit, view-based navigation [30,31,32,33,34].

Another question concerning the behaviour of dumpers is whether they head off in a similar direction from trip to trip, a behaviour called sector fidelity. In foraging, desert ants often show sector fidelity (*Cataglyphis bicolor*: [35,36,37]; *C. velox*: [38]; *M. bagoti:* [21,39]), as do some bull ants (*Myrmecia croslandi*: [40]; *Myrmecia midas*: personal observations), with the tendency increasing with experience [37]. Sector fidelity, however, depends on foraging success (*C. bicolor*: [41]). Colonies with lower foraging success or efficiency tend to exhibit lower sector fidelity. It is currently unknown whether dumpers show sector fidelity.

In this current study, we investigated the paths taken by both naive (inexperienced) and experienced ants when they engage in removing waste from inside the nest and depositing it away from the nest. Our initial hypothesis posited that naive dumpers would likely cover shorter distances compared to their experienced counterparts in the cluttered terrain. We also thought that naive ants would conduct at least one learning walk before dumping, as foragers conduct multiple learning walks before foraging [9], and nest excavators conduct one learning walk before excavating [24]. To understand these issues, we used videotaping to compare the paths of naive and experienced dumpers in their structure, duration, the distance traversed from the nest, travel speed, gaze-oscillation velocity, and scanning behaviour. We also examined the navigational capabilities of both naive and experienced *M. bagoti* dumpers using displacement tests. Both groups were tested at locations 2 m from the nest in four cardinal directions, with one of those directions being familiar to naive ants because their dumping trip was in that sector. We expected experienced dumpers to be well oriented in the home direction at all test sites, while naive dumpers would only be well oriented in the test location nearest their dumping spot. Additionally, we examined naive *M. bagoti* dumpers across their first five dumping trips, predicting that repeated trips would result in longer dumping distances and better navigational performance. This longitudinal design also allowed us to examine whether dumpers exhibit sector fidelity.

## 2. Methods

We conducted the experiments on three different colonies of the red honey ant *Melophorus bagoti* located on the outskirts of Alice Springs, Northern Territory, Australia (23°45.497′ S, 133°52.971′ E), in November 2022–March 2023 and November 2023–March 2024. One nest was used for each of three experiments. The semi-arid desert landscape surrounding the nest boasted predominantly buffel grass (*Pennisetum cenchroides*), interspersed with a mosaic of *Acacia* bushes and *Eucalyptus* trees [42]. In a radius of 7 m around the selected nest in Experiment 1, vegetation was removed before starting the investigation. In Experiment 2 and Experiment 3, we cleared the vegetation surrounding the tested *M. bagoti* nest within a radius of 10 m to improve the visibility of the dumpers’ activity. To enhance the visibility of the red ants against the red soil background, we spread fine white sand within the videorecording areas in all experiments. Our research adhered to ethical standards, with Australia lacking regulations pertaining to the study of ants. All experimental procedures employed throughout our investigation were entirely non-invasive, ensuring a conscientious approach to understanding the behaviour of *Melophorus bagoti* in its natural environment.

### 2.1. Animals

*Melophorus bagoti* colonies are widespread in the Central Australian semi-desert. They forage solitarily during the hot summer days, scavenging dead insects and collecting seeds, sugary plant exudates, and other miscellaneous items [21,22]. Our extensive fieldwork has revealed the monodomous nature of *M. bagoti* colonies in the region [42].

Our investigation required both naive and experienced dumping workers. The average foraging span of *M. bagoti* outside the nest is estimated at 4.9 days [21]. We took a 9-day period before beginning each of the experiments. For 9 continuous days, ants that exited from the nest were captured within a 30 cm radius using a wide-mouthed 50 mL glass container. They were then marked with black non-acrylic colour paint (Tamiya™, Boronia, Victoria, Australia) on the abdomen and released back into the nest, which is not harmful to ants. Beyond this period, any unpainted ants emerging with waste from the nest were almost certainly naive, with no experience of the outside world. During the study, we did not observe any identified ant that disappeared for 9 days or more and then reappeared outside, nor have we ever observed such a long gap of staying indoors in past research. We thus call such ants naive. These naive ants’ dumping activities were monitored until they performed over four days, making at least 5 dumping runs. After four days of experience outside, they were considered as experienced dumpers. In each experiment, we tested N = 15 ants for each group of experienced or naive dumpers. Each individual experienced or naive dumper was tested only once. To avoid repeated testing, ants were marked with an extra dot of paint on their thorax. To anticipate results, we found across three nests only four naive ants performing learning walks before engaging in dumping, while all other dumpers did not perform any learning walks. We painted these four a different colour to exclude them from the study.

### 2.2. Procedure

#### 2.2.1. Experiment 1

We followed dumpers during their outbound trip carrying nest-related wastes and inbound trips returning to the nest after they dumped nest-related wastes. We captured the inbound dumpers within 10 cm of the nest and displaced them to four test locations situated 2 m from the nest towards north, east, south and west (2 mN, 2 mE, 2 mS, 2 mW). We tested each dumper at these four locations in a random order. We chose to test only zero-vector ants to test for navigation based on views and not on path integration. Zero-vector ants have no vector to rely on. Test releases were close to the nest as past research suggests that after one trip outside, these ants are unlikely to be able to home from farther away [10]. Each test was videotaped using the set up and equipment described next in Experiment 2. Once the tests were completed, we marked the tested dumper with a yellow colour on the abdomen to avoid repeated testing, and then we released the tested ant back into the nest.

#### 2.2.2. Experiment 2

From the 10th morning onwards (following 9 days of painting emerging ants), we recorded all the outbound walks at a second nest using a video camera (Sony Handy camera FDR-AX700, Sony Australia, North Sydney, NSW, Australia) recording at a frame rate of 25 frames per second (fps) at a resolution of 3860 by 2160 pixels. A tripod was set at a height of 1.2 m from the ground, and the recording area measured 1 m by 1 m centred at the nest. The camera recorded every emerging ant, with an observer manually operating the ON/OFF of the recording button. The observer always stood next to the camera. When a dumping ant emerged carrying nest-related wastes of dead ants, foraged food, or empty cocoons, the observer let the camera continue recording until the ant returned to the nest to record their trip duration. Unpainted ants were naive and painted ants were experienced. Each ant was analyzed on only one trip. After recording the dumping ant’s duration and before the dumper entered the nest (within a 20 cm range), we turned off the camera, caught the dumper, and painted it with a different colour to avoid repeated observations of that individual ant. All the ants carrying experimental materials were followed individually, to document the distance at which ants dumped experimental materials. Thus, Experiment 2 measured the performance of dumpers on dumping trips, something not carried out in Experiment 1.

#### 2.2.3. Experiment 3

After 9 days, over 800 foragers were painted with a black dot of paint on their abdomen (Tamiya™). On the 10th day, any unpainted ant carrying waste material outside was considered a naive dumper. These individuals were caught at the nest entrance during their inbound run and marked with a unique pattern of paints (Tamiya™) on their body for consecutive trail observations. About 20 individual naive dumping ants were marked, and out of these, the first 15 ants that completed 5 dumping runs were selected for the experiment (N = 15). When these painted naive dumping ants emerged carrying nest-related wastes of dead ants, foraged food, or empty cocoons, the observer let the camera centred at the nest entrance continue recording until the ant returned to the nest vicinity to record its trip duration with a resolution of 3860 by 2160 pixels, using the same set up and equipment as in Experiment 2. Each dumper was analyzed for five consecutive trips. After recording the dumping ant’s duration on the first trip and before the naive dumper entered the nest (within a 20 cm range), we turned off the camera, caught the dumper, and painted it with a unique colour (Tamiya™) for individual identification. All the ants carrying waste materials were followed individually, to document the distance at which ants dumped waste materials. Thus, Experiment 3 followed the same ants over 5 runs, something not conducted in Experiments 1 and 2.

### 2.3. Tracking

From the videotapes, we used the deep learning animal-tracking programme DLTdv8 (version 8.2.9) in MATLAB (2022B) to extract frame-by-frame coordinates of the front of the head and middle of the thorax for individual dumpers obtained during our recordings [43]. These extracted frame-by-frame coordinates served as the basis for all analyses of the ants’ movements and behaviour.

### 2.4. Data Analysis

We analyzed the ants’ orientation and path characteristics while dumping. To understand dumping directional preference, we used the circular statistics package in R (version 4.2.1; R Core Team, 2020). On videotaped data, we measured exit direction as the vector from the thorax to the head in the final frame on the record in which the ant was visible. In Experiment 3, we also measured a final heading direction, defined by the direction to the point at which the waste was dumped.

To understand how quickly the ants moved, we calculated speed. Speed refers to the magnitude of an ant’s velocity and was calculated as the average over the entire videotaped trajectory for each ant, excluding the stopping durations. We also calculated the orientation angular velocity by measuring the rate of change in an ant’s orientation direction over time. When walking, the gaze direction of these ants typically oscillates continuously. Orientation direction in any frame was determined as the direction of the straight line passing from the thorax coordinates through the head coordinates. Orientation angular velocity was calculated by dividing the change in orientation angle by the corresponding change in time, providing a measure of how quickly the ant is altering its viewing direction.

To understand path measures, we used three indices of straightness, path straightness, sinuosity, and $E_{max}^{a}$, each of which relates to the directness of navigation away from the nest. Firstly, straightness is computed as the ratio of the straight-line distance between the starting point (nest) and final point (the last frame before an ant exited the recording area) of the path to the overall length of the path [44,45,46,47,48,49]. Straightness ranges from 0 to 1, with larger values indicating straighter paths, while smaller values indicate more curved or convoluted paths. Sinuosity is an estimate of the tortuosity in a path, calculated as $S=2{\left[p\left(\frac{1+c}{1-c}+b^{2}\right)\right]}^{-0.5}$, where p is the mean step length, c is the mean cosine of turning angles and b is the coefficient of variation of the step length. Step lengths are the Euclidean distances between consecutive points of the thorax along a path. The turning angle refers to the change in direction between two consecutive steps. Sinuosity varies between 0 (straight) and 1 (extremely curved) [50]. The maximum expected displacement of a path, $E_{max}^{a}=\frac{\beta }{1-\beta }$, where β is the mean cosine of turning angles, is a dimensionless value expressed as a function of number of steps, and is consistent with the intuitive meaning of straightness [51]. Larger maximum expected-displacement values indicate straighter paths and hence greater displacement, while a smaller value suggests more localized or constrained movement. Paths were characterized and visualized in R (version 4.2.1; R Core Team, 2020) using the packages trajr [52] and Durga [53].

During the outbound navigation, ants frequently displayed a series of stereotypical successive fixations in different head directions by rotating on the spot at one location, known as a “scanning bout” [54]. During each dumper’s run, we extracted the number of scanning bouts and the scanning-bout durations (from the start of a scanning bout until the ant started walking again).

To measure the distance and duration of both naive and experienced dumpers during their runs, we implemented a systematic procedure. We noted the starting point of each ant’s journey as it left the nest until it arrived within 20 cm of their nest and considered that duration as their trip duration. During the outbound run we marked the maximum distance as the endpoint of their trip, with the waste always being dumped at this farthest point. The distance travelled was calculated by measuring the linear distance between the starting point at the nest and the marked endpoint.

### 2.5. Statistical Analysis

#### 2.5.1. Experiment 1

We performed a Rayleigh test to assess whether the distribution of headings (exit directions) for each test location was uniform (*p* > 0.05). For the naive ants, we tested an additional heading distribution consisting of the test performance of each ant at the test site nearest to where the ant dumped its waste. This distribution was not compared with any other distribution. For the other distributions, in case two distributions turned out non-uniform, we planned to compare the mean direction of the two groups using the Watson–Williams test (alpha = 0.05). If the hypothesis of a uniform distribution cannot be rejected for one or both groups, it makes no sense to run this test. Additionally, we examined whether final heading orientations significantly clustered around the nest direction at 0 degrees by checking whether 0 degrees fell within the 95% confidence interval (CI) of orientations (Watson tests). V-tests were also conducted, with alpha set at alpha = 0.05, to determine if the mean headings were notably clustered around the nest direction. Single-sample log likelihood ratio tests were also conducted to investigate whether the heading distributions of the ants were uniform in each test condition.

We used generalized linear mixed models to investigate whether the displacement location affected dumpers’ path straightness, sinuosity, $E_{max}^{a}$, speed and orientation angular velocity. The model treated dumper type (experienced or naive) as a between-subject factor and displacement locations as a within-subject variable that predicted the dependent variables. The LM was formulated using the lme4 package (version 1.1-27) and fitted using the lmer function Since path straightness and sinuosity are bounded measures (0–1), we used the binomial family, whereas all of our other response variables are only bounded by zero, and so for them, we applied the Gaussian family of models. We used the Tukey post hoc test to perform pair-wise post hoc comparisons for each of our models. Since we tested multiple dependent variables computed from the same data sets of trajectories on tests, we adopted alpha = 0.01 to lower type 1 errors. Statistical analyses were conducted using R (version 4.3.1).

#### 2.5.2. Experiment 2

To assess the uniform distribution of exit directions for each group (*p* > 0.05), we performed a Rayleigh test. In case both distributions turned out non-uniform, we planned to compare the mean direction of the two groups using the Watson–Williams test (alpha = 0.05). In evaluating the navigation routes of experienced and naive dumpers, our models compared path characteristics. Ant experience—experienced or naive—was treated as an independent variable (X) that predicts some dependent variable (Y). We used 4 individual models to compare the effect of the displacement location on path straightness, sinuosity, $E_{max}^{a}$, and speed. We used 3 individual models to compare the number of scanning bouts per test and the mean durations of scanning bouts, as well as the dumping distance. We conducted a Welch’s ANOVA (one-way) on scanning duration, an analysis that is suitable for data with significant heterogeneity of variance, which emerged for that variable with Levene’s test. The GLM was formulated using the lme4 package (version 1.1-27) and fitted using the glmer function. We again used the binomial family for path straightness and sinuosity and the Gaussian family of models for all other dependent variables. Alpha was set at 0.01 again to lower type 1 errors. Statistical analyses were conducted using R (version 4.3.1). We visualized the data using summary characteristics such as median (in box plots), density and mean confidence intervals (in violin plots).

#### 2.5.3. Experiment 3

For characteristics of paths, we compared variables across the 5 trials using a linear-model ANOVA (alpha = 0.01); to compare trials, we applied a priori Helmert contrasts. These independent contrasts were set for Trip1 vs. Trip2 to Trip5 (Contrast-1), Trip2 vs. Trip3 to Trip5 (Contrast-2), Trip3 vs. Trip4 and Trip5 (Contrast-3), and Trip4 vs. Trip5 (Contrast-4). Trials formed the only within-subject independent variable, and ants formed a random factor. Our hypothesis before data analysis was that only the first and second Helmert contrast would reach significance.

For circular data on exit directions, we again checked whether a distribution of headings was uniform using a Rayleigh test. The mean directions of corresponding distributions were compared with the Kuiper test, Watson–Williams multi-sample test, and Hotelling’s test.

## 3. Results

### 3.1. Experiment 1

The majority of naive dumpers did not engage in a learning walk before heading off on their first dumping job: 15 did not while 4 did. Of the latter, three took two learning walks while one took a single learning walk. As indicated in the Methods, we excluded the entire latter cohort and studied only the naive dumpers that did not perform a learning walk. All the dumpers adhered to a consistent procedure: they carried an item using their mandibles, adjusting its position to hold it off the ground, walked with the object for some distance, and then discarded it with a characteristic lunge. Sometimes, they performed a number of scanning bouts on their way out while holding the waste item. Our informal observations on experienced dumpers suggested that most of these workers stuck to their job, undergoing a period of intensive dumping activities typically spanning 3 to 4 days. They would have taken many trips >5 m in distance (based on earlier data in [9]) when they were tested in this study. None were observed to take learning walks.

In order to understand the navigational knowledge of these experienced and naive dumpers, after their dumping activity, each ant was captured just before it entered the nest and displaced to 2 m from the nest in four different locations to the North, East, South and West. The ants’ final headings at each location showed that the experienced dumpers in the 2 m displacements were oriented towards the nest direction of 0 deg from all four cardinal directions. In contrast, naive dumpers that were displaced 2 m from the nest were not oriented towards the nest direction at any release point (Figure 1). Based on the Rayleigh test, the experienced dumpers’ initial orientations were non-uniformly distributed in all 2 m displacement tests, whereas the naive dumpers were scattered, with uniformly distributed headings (Table 1, Figure 1). In addition, experienced dumpers in 2 m displacement conditions showed significant V-test results in the nest direction, and the means of their 95% confidence interval of initial heading values include the nest direction of 0 deg (Watson test, *p* > 0.05). The log likelihood ratio test for the experienced dumpers on 2 m tests failed to reject the hypothesis that the distribution was clustered in the home direction (*p* ≥ 0.05, k ≥ 0, χ^2^ ≤ 1). For the naive dumpers on 2 m displacement tests, however, the log likelihood ratio rejected the hypothesis that the mean value of distribution was equal to the predicted value (home direction) (*p* = 0.0006, k = 0.64 and χ^2^ = 5.6), meaning that the headings were oriented in a different direction from the nest direction. We checked whether the naive dumpers were well oriented at the test location nearest to their dumping point. The majority of the naive dumpers oriented in the general direction of the nest (Table 1 and Figure 1I). Both the Rayleigh test and the V-test returned significant values, indicating an oriented distribution in the nest direction.

We checked whether experienced dumpers differed from naive dumpers in their path characteristics at the four different displacement locations. Firstly, the experienced dumpers exhibited lower tortuosity in their paths, indicating that they changed their travelling direction fewer times compared to the naive dumpers, which tended not to take direct paths. The linear mixed-model ANOVA showed significant differences in sinuosity between the experienced and naive groups (F_1, 59_ = 33.64, *p* < 0.0005) (Figure 2A). However, no significant difference was found within the experienced and naive groups across test locations (F_3, 57_ = 0.17, *p* = 0.91). Additionally, the model did not detect any significant interaction (F_3, 57_ = 0.44, *p* = 0.71). Overall, the sinuosity measure confirmed that the experienced dumpers showed more efficiency in paths than did the naive dumpers. Secondly, in $E_{max}^{a}$, the degree of displacement varied between the experienced and naive dumpers. The linear mixed-model ANOVA showed significant differences in $E_{max}^{a}$ between the experienced and naive groups (F_1, 59_ = 32.29, *p* < 0.0005) (Figure 2B). However, no significant difference was found within the experienced and naive groups across locations (F_3, 57_ = 1.2, *p* = 0.29). Additionally, the model did not detect any significant interaction (F_3, 57_ = 0.18, *p* = 0.9). Finally, with straightness, the analysis of variance found statistical significance only between groups (F_1, 59_ = 55.9, *p* < 0.0005) (Figure 2C), with a trend across test locations (F_3, 57_ = 3.52, *p* < 0.018) and in the interaction (F_3, 57_ = 2.79, *p* < 0.04).

During the displacement test, we found that the lack of experience in naive dumpers had a noticeable impact on the speed of the ants (Figure 3A,B). The linear mixed-model ANOVA showed significant differences in mean speed between the experienced and naive groups (F_1, 59_ = 31.94, *p* < 0.0005) (Figure 3A). However, no significant difference was found within the experienced and naive groups across locations (F_1, 59_ = 0.82, *p* = 0.48). Additionally, the model did not detect any significant interaction (F_3, 57_ = 1.8, *p* = 0.14). As for orientation angular velocity, the linear mixed-model ANOVA showed a trend between the experienced and naive groups (F_1, 59_ = 5.5, *p* < 0.02) (Figure 3B). However, no significant difference was found within the experienced and naive groups across locations (F_3, 57_ = 0.82, *p* = 0.47). Additionally, the model did not detect any significant interaction (F_3, 57_ = 2.1, *p* = 0.1).

During the displacement tests, we found that naive dumpers showed more scanning behaviour compared to experienced dumpers (Figure 4A,B). The linear mixed-model ANOVA showed significant differences in mean number of scanning bouts between the experienced and naive groups (F_1, 59_ = 72.95, *p* < 0.0002) (Figure 4A). However, no significant difference was found within the experienced and naive groups across displacement locations (F_3, 57_ = 0.74, *p* = 0.52). Additionally, the model did not detect any significant interaction (F_3, 57_ = 0.61, *p* = 0.60). Similarly, with the scanning-bout durations, the linear mixed-model ANOVA showed significant differences between the experienced and naive groups (F_1, 59_ = 9.23, *p* < 0.005) (Figure 4B). Nevertheless, no significant difference was found within the experienced and naive groups across displacement locations (F_3, 57_ = 0.2, *p* = 0.89) and the model did not detect any significant interaction (F_3, 57_ = 2.45, *p* = 0.06).

### 3.2. Experiment 2 and Experiment 3

We observed individual experienced and naive dumpers removing waste items from inside the nest, dumping the waste outside the nest, and then returning home. We observed even fewer learning walks than in Experiment 1: none of the naive dumpers took a learning walk before going off dumping. However, some of the naive dumpers performed relearning walks after their first dumping activity and then switched their roles to excavators. Data from these dumpers were removed from this study.

Experience had a noticeable impact on the speed of the ants. In Experiment 2, the experienced dumpers moved faster from the nest (81.56 mm/s) compared to the naive dumpers (49.97 mm/s) (Figure 5A,B). The generalized linear-model ANOVA showed a significant difference between the experienced and naive dumpers (Z_1, 28_ = 4.52, *p* = 0.0001). In the orientation angular velocity of gaze-direction change, the experienced dumpers showed a numerically higher orientation angular velocity (246.9 deg/s) than the naive dumpers (210.8 deg/s) (Figure 5B). This difference, however, was not significant statistically (Z_1, 28_ = –1.2, *p* = 0.5). In Experiment 3, we found a notable difference in mean speed across trials of dumping trips. The mean speed during the initial dumping trips was significantly slower than during the other four dumping trips (Figure 5C, a priori Helmert Contrast-1: Z_1, 70_ = 3.15, *p* = 0.006). Contrast-2 revealed a trend (Z_1, 70_ = 2.61, *p* = 0.026), while other Helmert contrasts revealed no significant differences between the trials (in Contrast-3, Z_1, 70_ = 1.37, *p* = 0.3; in Contrast-4, Z_1, 70_ = –0.78, *p* = 0.43). In orientation angular velocity of gaze-direction change, no significant differences across trials was found (Figure 5D; Z_4, 70_ = 0.89, *p* = 0.37).

Experienced dumpers appeared to walk in a straighter fashion, but statistically significant differences were found only in sinuosity and straightness (Figure 6). In sinuosity, the generalized linear-model ANOVA in Experiment 2 showed a significant difference between the groups (Z_1, 28_ = 4.11, *p* = 0.0003) (Figure 6A). In $E_{max}^{a}$, on the other hand, the generalized linear-model ANOVA did not show a significant difference between the groups (Z_1, 28_ = 1.53, *p* = 0.23), (Figure 6B). Finally, in straightness, the experienced dumpers were significantly straighter (Z_1, 28_ = 13.56, *p* = 0.0001) (Figure 6C). In Experiment 3, we found a notable difference in sinuosity across trials of dumping trips. The paths were more tortuous, wigglier, and significantly less straight during the initial dumping trips than during the fifth dumping trip (Figure 6D). The a priori Helmert Contrast-1 (Z_1, 70_ = 3.02, *p* = 0.007) and Contrast-2 (Z_1, 70_ = 3.63, *p* = 0.001) revealed significant effects, while Contrast-3 (Z_1, 70_ = 2.53, *p* = 0.02) revealed a trend. No significant effect was found in Contrast-4 (Z_1, 70_ = 0.2, *p* = 0.81). $E_{max}^{a}$ differed significantly only in Contrast-1 (Z_1, 70_ = 3.24, *p* = 0.002) and Contrast-2 (Z_1, 70_ = 3.65, *p* = 0.001), with other Helmert contrasts revealing no significant effects (Contrast-3, Z_1, 70_ = 1.68, *p* = 0.2; Contrast-4, Z_1, 70_ = –0.23, *p* = 0.81). For straightness, the generalized linear-model ANOVA showed no significant differences across trips (Z_4, 70_  =  1.59, *p*  =  0.05).

Naive dumpers showed more scanning behaviour than experienced dumpers. In Experiment 2, all the naive dumpers (15/15) performed at least two scanning bouts during navigation from the nest, whereas only a minority of experienced dumpers (6/15) performed any scanning bout at all in the recording area (Figure 7A). This difference in number of scanning bouts is statistically significant (Z_1, 28_ = 13.12, *p* = 0.005). Durations of scanning bouts were shorter in experienced ants, but the difference just failed to be significant (Welch’s one way ANOVA, F_1, 28_ = 8.08, *p* = 0.012) (Figure 7B). In scans in Experiment 3, the analysis indicated significant changes in the number across trips. The generalized linear-model ANOVA revealed a significant effect of trips on scans (Z_4, 70_ = 60.748, *p* < 0.001), with decreases in scans between Trip1 and all subsequent trips, suggesting that the number of scans decreased significantly in the later trials (Figure 7C). The a priori Helmert Contrast-1 (Z_1,70_ = 3.12, *p* = 0.006) and Contrast-2 (Z_1, 70_ = 3.03, *p* = 0.008) revealed significant effects. However, other Helmert contrasts revealed no significant effects (Contrast-3, Z_1, 70_ = 2.33, *p* = 0.02; Contrast-4, Z_1, 70_ = 0.2, *p* = 0.81). Significant changes in durations of scans across trips were also observed. The generalized linear-model ANOVA revealed a significant effect of trips on scan duration (Z_4, 70_ = 27.92, *p* < 0.001). The a priori Helmert Contrast-1 (Z_1,70_ = 3.09, *p* = 0.007) and Contrast-2 (Z_1, 70_ = 3.14, *p* = 0.001) revealed significant effects. However, other Helmert contrasts revealed no significant effects (Contrast-3, Z_1, 70_ = –0.53, *p* = 0.22; Contrast-4, Z_1, 70_ = 0.2, *p* = 0.81) (Figure 7D).

The time taken to return to the nest and the distance travelled by these dumpers showed significant differences between experienced and naive dumpers in Experiment 2, with experienced ants being longer in both variables. The generalized linear-model ANOVA showed a significant difference between conditions in duration (Z_1, 28_ = 28.17, *p* = 0.0001) (Figure 8A) and distance (Z_1, 28_ = 29.04, *p* = 0.0001) (Figure 8B). In Experiment 3, the generalized linear-model ANOVA demonstrated significant increases across trips in duration (Z_4, 70_ = 27.92, *p* < 0.001) and distance (Z_4, 70_ = 25.62, *p* < 0.001) (Figure 8). With duration, the a priori Helmert Contrast-1 (Z_1,70_ = 3.55 *p* = 0.001) revealed significant effects, while Contrast-2 (Z_1, 70_ = 2.47, *p* = 0.03) revealed a trend. However, other Helmert contrasts revealed no significant effects (Contrast-3, Z_1, 70_ = –1.63, *p* = 0.19; Contrast-4, Z_1, 70_ = –0.52, *p* = 0.59) (Figure 8C). With distance, the a priori Helmert Contrast-1 (Z_1,70_ = 3.55, *p* = 0.001) and Contrast-2 (Z_1, 70_ = 3.03, *p* = 0.006) revealed significant effects, while Contrast-3 (Z_1, 70_ = 2.47, *p* = 0.03) revealed a trend. No significant effect was found in Contrast-4 (Z_1, 70_ = −0.67, *p* = 0.59) (Figure 8D).

To understand whether experienced and naive dumpers have specific directional preferences, we examined their heading directions from the nest. In Experiment 2, the final headings of both experienced (mean vector direction (µ) = 141.26, length of mean vector (r) = 0.37) and naive dumpers (mean vector direction (µ) = 42.99, length of mean vector (r) = 0.28) were randomly distributed around the nest (Appendix A Figure A1). The Rayleigh test revealed that the ants’ initial and final heading orientations were uniformly distributed in experienced (Z = 2.07, *p* = 0. 12) and naive dumpers (Z = 1.22 *p* = 0.3) (Appendix A Figure A1). This suggests that dumpers as a population do not head off in a preferred direction or sector. In heading direction across all dumpers in Experiment 3, the distribution of directions from the nest was random (Appendix A Figure A1). The Rayleigh test revealed that the ants’ final heading orientations were uniformly distributed in all dumping trials (Trip1: Z = 2.78, *p* = 0. 06; Trip2: Z = 0.36, *p* = 0. 7; Trip3: Z = 1.23, *p* = 0. 29; Trip4: Z = 1.05, *p* = 0. 35; Trip5: Z = 1.77, *p* = 0. 17). We also compared the empirical distributions across the five trips with Kuiper and Watson–Wheeler tests, and both showed non-significance in direction heading (Kuiper: V = 0.13.4, *p* = 0.65; Watson–Wheeler test: W = 2.61, *p* = 0. 26). This means that the five trips were not significantly different in mean or variance. Dumpers as a population did not head off in a preferred direction. To examine sector fidelity in individual ants across trips (Figure 9), we conducted Hotelling’s test, and it showed significant directional preference in individuals (F = 60.46, *p* = 0.004). Dumpers as a group showed sector fidelity.

## 4. Discussion

Our study on the Australian red honey ant, *Melophorus bagoti,* highlighted significant differences in the navigational abilities of naive and experienced dumpers. Compared to experienced dumpers in Experiment 1, naive dumpers were not oriented towards the nest from 2 m away, exhibited more meandering behaviour in tests, and walked more slowly. They scanned their surroundings more frequently and for longer durations, indicating less navigational knowledge. However, naive dumpers showed some nestward orientation at locations nearest their dumping site, suggesting some learning. Experiment 2 found that colony waste was deposited in random directions, with naive ants not taking learning walks before their first dumping trip. On dumping trips, experienced dumpers walked faster, changed head direction more quickly, and dumped waste at greater distances. Naive dumpers meandered more, walked with more wiggling, and took shorter, more erratic trips, scanning more often than experienced dumpers. In Experiment 3, dumpers also did not take learning walks before their dumping tasks but showed sector fidelity, mostly taking waste in the same general direction from trip to trip. Their navigational performance improved over their first five trips, walking faster, taking waste farther, scanning less, and exhibiting more directed movement with lower sinuosity and higher straightness in travel. These findings suggest that dumpers learn and improve their efficiency over time.

Why did such differences in dumping behaviour emerge? In one word, our interpretation is learning. Learning led to differences in two ways. First, experienced ants have learned to navigate well in the area around the nest, or at least in the direction in which they are carrying waste. They can thus execute their dumps by travelling faster and in reasonably straight paths with little scanning, carrying the waste a much longer—and for the nest, safer—distance before dumping the possibly contaminated and pathogenic waste [9]. In the distance of dumping, the two groups show no overlap at all in their data (Figure 7B). Second, naive dumpers were ‘learning on the job’, combining a learning walk with dumping to ensure that they could find their way back. The slower movement, the more tortuous paths, and the scanning are all reminiscent of the learning walks in this and other species ([10,55,56], reviews: [57,58]) as well as the behaviour of ants when the navigational situation is unfamiliar or has been changed [46,47,48,59,60,61]. Potential foragers usually perform a number of learning walks before going off foraging. And although *M. bagoti* foragers are efficacious at returning from an experimental feeder on their first visit, they take a number of trips to venture to a feeder 8 m from the nest [57]. For foragers, the learning walks facilitate returning home from some distance [10,32,61,62]. Thus, we think that some learning on the job is necessary for naive dumpers to return home successfully and yet still from much shorter dumping distances than those of the trips of experienced foragers.

Our learning interpretation, however, raises another intriguing question: if learning is needed for dumping waste, why do naive dumpers not carry out learning walks before dumping? Even excavators, those workers dropping sand from the nest just outside the nest, take one learning walk resembling the first learning walk of future foragers before excavating [24]. Sand is dumped within 15 cm of the nest [9,24], an order of magnitude shorter in distance than the distances at which naive dumpers in this study carried their waste. Although more speculative on this question, we suspect that the functional answer lies in the potential pathogenicity of the material being dumped. From the perspective of the nest as a whole, sand can wait for excavators to learn their surrounds, but dead ants and possibly rotting food should not wait. The ‘learning-on-the-job’ strategy facilitates removing the waste from the nest as soon as possible. More research is needed, however, to answer this question convincingly.

Our studies on dumping ([9]; current results), in contrast to the findings of learning walks in excavators [24], raise another, mechanistic question: on what basis do the ants decide to dump an item without first taking a learning walk? Examining the extant literature, we suspect that chemicals are at play as triggering cues, which emanate from dead arthropods, which comprise the waste that *M. bagoti* dumpers carry from their nest. The chemicals that have been shown experimentally to trigger removal behaviour in ants are oleic acid (*Pogonomyrmex badius* and *Solepnopsis saevissima*: [63]; *Myrmeica vindex*: [64]; *Myrmica rubra*: [65]; *Solepnopsis invicta*: [66]) and linoleic acid (*M. rubra*: [67]; *S. invicta*: [66]; review: [68]), a monounsaturated and polyunsaturated fatty acid, respectively. Both these fatty acids increase in a corpse with time since death [65]. Other cues could add potency in triggering removal behaviour, such as infection in the corpse; compared with corpses that have died from being frozen, corpses that died from fungal infection elicit stronger behaviours (*M. rubra*: [67]). Fungal infection, however, results in more oleic and linoleic acid in the corpse one day after death (*S. invicta*: [65]), so this additional cue might be capitalizing on the fatty-acid triggering route. Waste removal behaviour is enhanced when a colony has brood, so other signals than those from fatty acids likely also act as modulators or cues for waste removal (*M. rubra*: [69]). One gene has been identified as crucial for such a triggering effect of oleic and linoleic acids: the chemosensory protein gene *Si-CSP1* [70].

When it comes to mechanisms underlying behaviour, the central complex of the insect brain is crucial for coordinating behaviours in navigation [71,72]. We suspect that this neural region plays a crucial role in integrating inputs and generating outputs in dumping as well. A vector of some magnitude is likely an output. In both *M. rubra* [73] and *M. bagoti* (current results), the heading direction of each ant is idiosyncratic, making a uniform distribution across the dumping population. All this suggests that a code of distance determines how far to take the waste in a favoured sector. We have suggested that both path integration and view learning contribute to navigation in dumpers [9]. Given only a little view learning on the first dumping run and the inexactitude of path integration, naive ants might reduce risk by opting for shorter travel distances compared to experienced dumpers. The frequent scanning behaviour could also serve to compensate for path-integration-vector errors, enabling a multi-cue integration of path integration and scanning behaviour to guide dumpers back to the nest [74,75]. The cognitive factors that drive decisions to dump at different distances across dumping trips, however, require more investigation, and between the inputs that trigger expression of the *Si-CSP1* gene and the vector output, neurobiologists have much to investigate to trace the connections.

Our study found sector fidelity in dumping: most dumpers headed in the same general direction trip after trip (Figure 9). Desert ant foragers also exhibit sector fidelity, consistently foraging in a particular direction across multiple trips [21,38,39,41]. In *Cataglyphis bicolor*, sector fidelity improved foraging success, although foragers may switch to a neighbouring sector with repeated failures [41]. Sector fidelity might begin before foraging. In red honey ants, the centroid direction of the last learning walk of individual ants predicts the direction of their first foraging trip [10]. Future foragers might prioritize learning views from specific directions rather than focusing on panoramas from all directions. We think that dumpers have even stronger reasons for exhibiting sector fidelity. Firstly, without prior learning walks, the results of Experiment 1 suggest that dumpers probably learned to home from only one sector. Sticking to the same sector would allow them to encounter more familiar views, facilitating learning, which in turn results in the more efficient job performance that develops over trips, as found in this study. Secondly, unlike a foraging sector, a dumping sector does not offer cases of failure—as long as the dumper gets home again—because each sector contains plenty of space for dumping waste. A dumper is unlikely to find a reason for switching sectors.

In conclusion, our study on dumpers of the Australian red honey ant *Melophorus bagoti* highlights significant differences between naive and experienced dumpers. Naive dumpers did not demonstrate orientation towards the nest from 2 m away, whereas experienced dumpers did. Furthermore, compared to experienced counterparts, naive dumpers walked slower and displayed more meandering behaviour with increased sinuosity and reduced straightness. Almost none of the naive dumpers took a learning walk before their first dumping trip. They appeared to be learning on the job. With repeated runs, however, these ants improved their navigational performance, walking longer distances to dump waste, and travelling at faster speeds, executing straighter paths with fewer scanning bouts. This progression suggests that the ants gradually acquire and refine their spatial knowledge over successive trips, while developing a preference for consistent dumping sectors. This sector fidelity parallels the behaviours found in desert ant foragers, which also demonstrate sector fidelity to maximize foraging success and navigational efficiency. Sector fidelity might be a common adaptive trait in ant species to optimize task performance and survival. All in all, the findings underscore the importance of learning and experience in shaping the navigational knowledge of desert ants.

## Figures and Tables

**Figure 1 insects-15-00814-f001:**
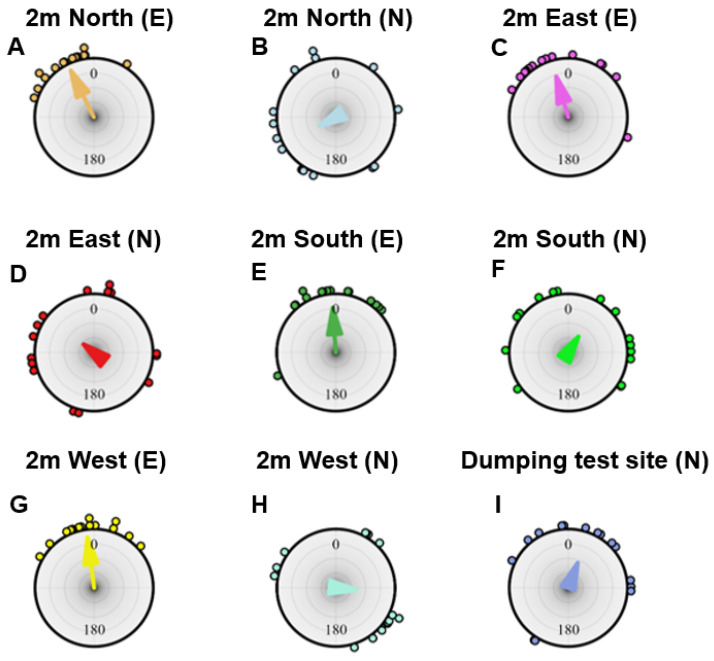
Headings on displacement tests. Circular histograms of initial headings of foragers during the displacement tests at 2 m north in experienced (**A**) and naive (**B**) ants, at 2 m east in experienced (**C**) and naive (**D**) ants, at 2 m south in experienced (**E**) and naive (**F**) ants, and at 2 m west in experienced (**G**) and naive (**H**) ants, and at the test site nearest to the dumping location of each naive ant (**I**). In the figure title, E denotes experienced and N denotes naive. In the histograms, the nest direction is set at 0°. The arrows denote the length and direction of the mean vector.

**Figure 2 insects-15-00814-f002:**
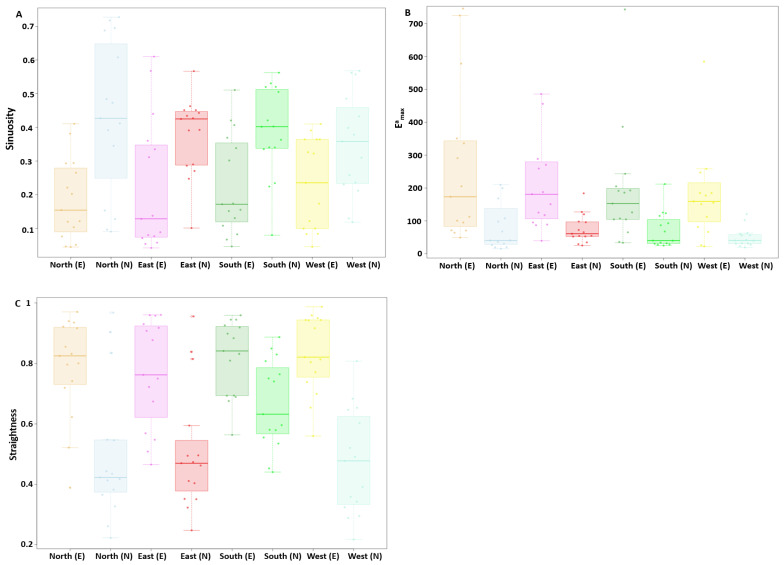
Path characteristics of dumpers at various test locations. (**A**) Sinuosity, (**B**) $E_{max}^{a}$ and (**C**) straightness. Box plots display the median (line inside the box), interquartile range (box), and extreme values excluding outliers (whiskers). Individual data points are shown as dots. In the *x* axis of the figure legend, “E” after the direction name denotes experienced, whereas “N” denotes naive.

**Figure 3 insects-15-00814-f003:**
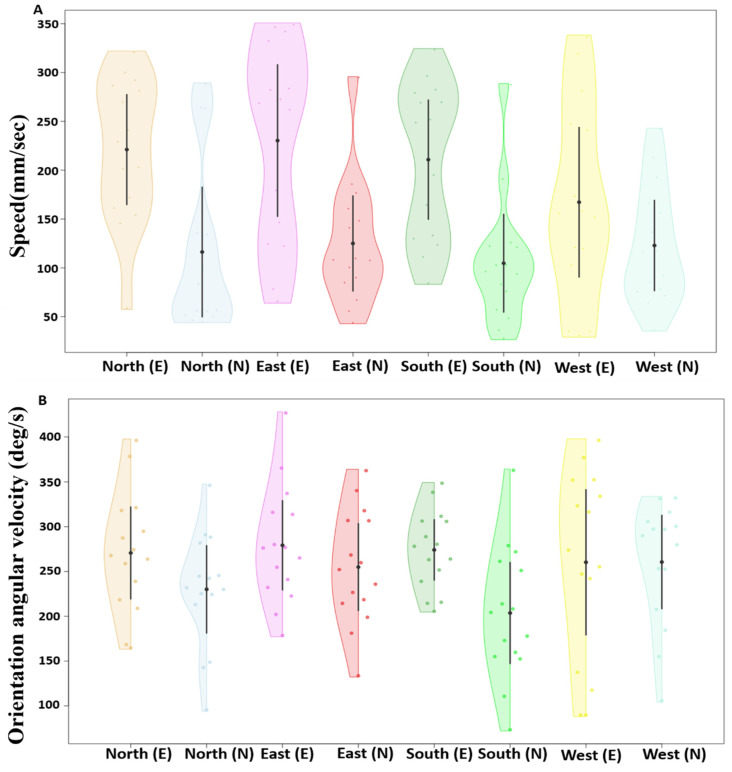
Speed and orientation angular velocity of displaced ants at different test locations. The violin plots show the mean speed of ants across their entire trajectory at each displacement location (**A**). The half-violin plots show the distribution of bootstrapped differences in mean orientation angular velocity of ants at the various displacement locations (**B**). In (**A**,**B**), the solid dot shows the mean, while the vertical bar shows the 95% confidence interval of the mean. In the *x* axis figure legend, “E” after the direction name denotes experienced, whereas “N” denotes naive dumpers.

**Figure 4 insects-15-00814-f004:**
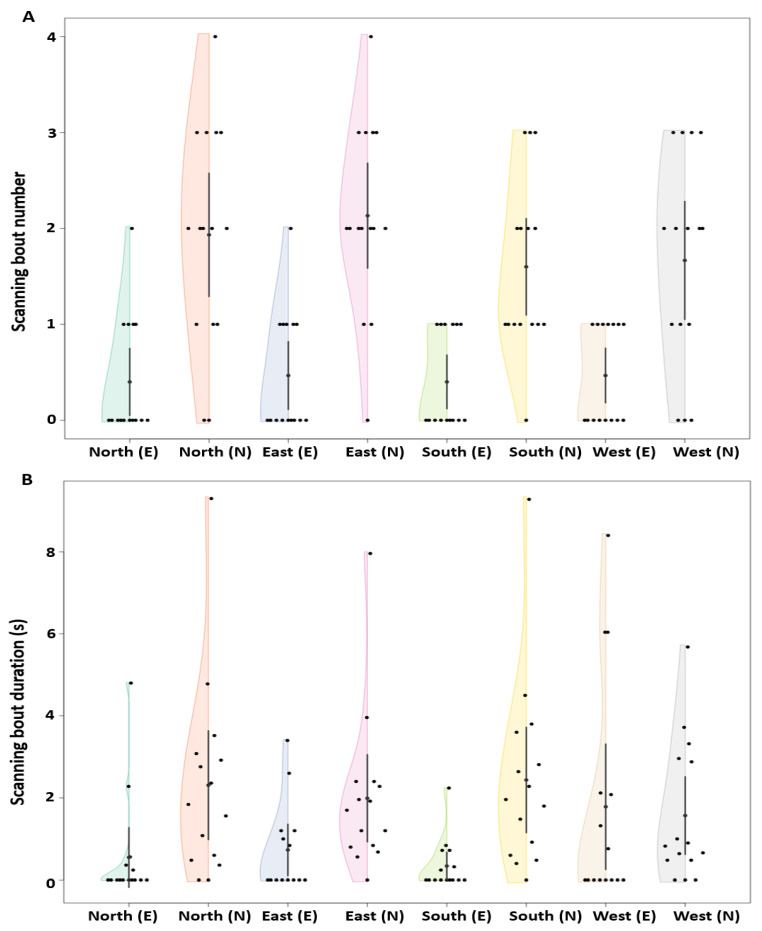
Number and timing of scanning bouts. Number of scanning bouts (**A**) and scanning-bout duration (**B**) of the experienced and naive dumpers during the displacement tests. The half violins show the distribution of bootstrapped differences; the solid dot shows the mean, while the vertical bar shows 95% confidence interval of the mean. In the *x* axis figure legend, “E” after the direction name denotes experienced, whereas “N” denotes naive dumpers.

**Figure 5 insects-15-00814-f005:**
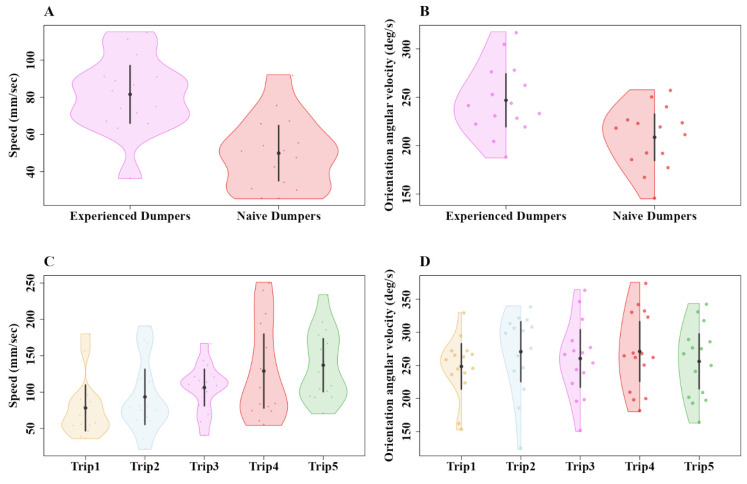
Comparison of speed and orientation angular velocity of experienced and naive dumpers in Experiment 2 and over five consecutive dumping trips in Experiment 3. The violin plots show the mean speed of experienced and naive dumpers across their entire trajectory in Experiment 2 (**A**) and the mean speed across trials in Experiment 3 (**C**). The half-violin plots show the distribution of bootstrapped differences in mean orientation angular velocity of experienced and naive dumpers in Experiment 2 (**B**) and across trials in Experiment 3 (**D**). The solid dots show the mean, while the vertical bars show the 95% confidence interval of the mean.

**Figure 6 insects-15-00814-f006:**
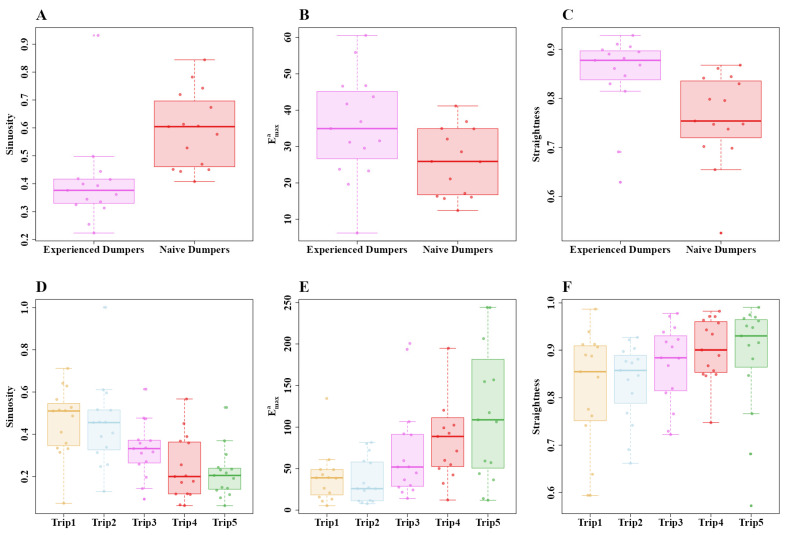
Path characteristics of experienced and naive dumpers in Experiment 2 (**A**–**C**) and across the five consecutive dumping trips in Experiment 3 (**D**–**F**). (**A**,**D**) Sinuosity, (**B**,**E**) *E_max_* and (**C**,**F**) straightness. Box plots display the median (line inside the box), interquartile range (box), and extreme values excluding outliers. Individual data points are shown as dots.

**Figure 7 insects-15-00814-f007:**
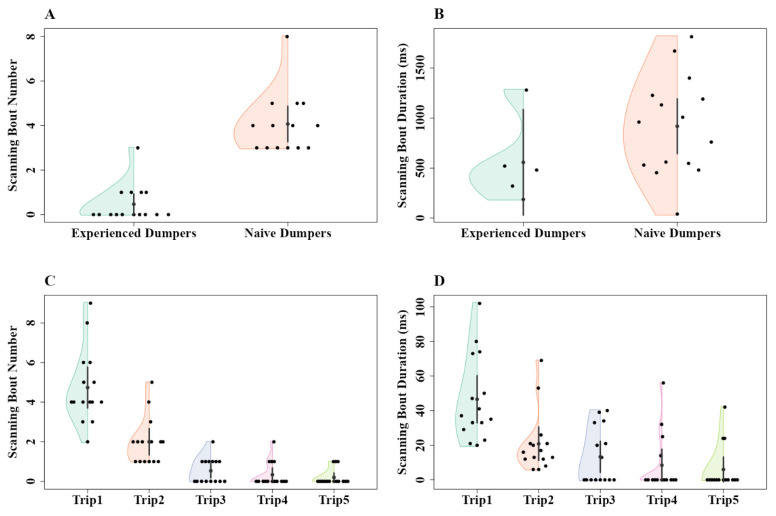
The number of scanning bouts and duration of scanning bouts of experienced and naive dumpers in Experiment 2 (**A**,**B**) and across five trips in Experiment 3 (**C**,**D**). Number of scanning bouts (**A**,**C**) and scanning-bout duration (**B**,**D**). The half violins show the distribution of bootstrapped differences; the solid dot shows the mean, while the vertical bar shows the 95% confidence interval of the mean.

**Figure 8 insects-15-00814-f008:**
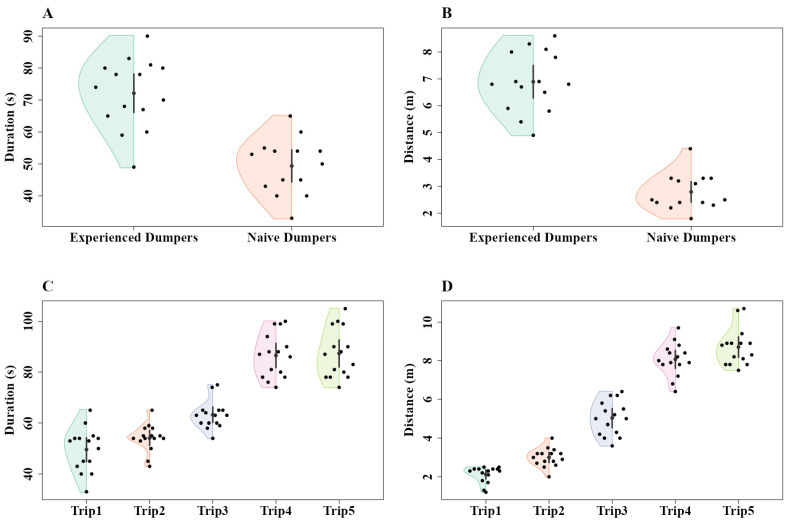
Comparison of dumping walk duration and waste disposal distance between experienced and naive dumpers in Experiment 2 (**A**,**B**) and over five consecutive trips in Experiment 3 (**C**,**D**). The duration of the entire run (**A**,**C**) and distance at which waste was dumped (**B**,**D**) in the dumping runs. The half violins show the distribution of bootstrapped differences; the solid dot shows the mean, while the vertical bar shows the 95% confidence interval of the mean.

**Figure 9 insects-15-00814-f009:**
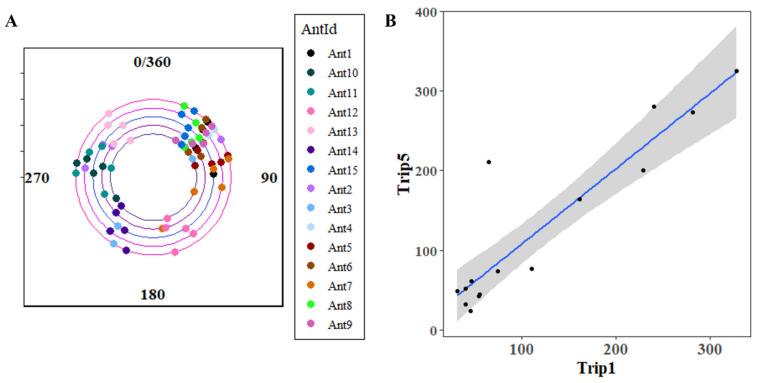
Heading directions across the 5 dumping walks in Experiment 3. South is at 0/360. The circular correlogram shows the final heading direction of the ants from Trip1 to Trip5 (**A**). The inner circle shows Trip1 and the outer circle shows Trip5. (**B**) The heading directions on the first and fifth trips of dumpers. The right edge (360 deg) wraps around to the left edge (0 deg) and the top edge (360 deg) wraps around to the bottom edge (0 deg). It can be seen that 13 of 15 points (ants) fall near the diagonal line, which indicates the line of equal final heading directions on the first and fifth walks. The grey bar shows the 95% confidence interval.

**Table 1 insects-15-00814-t001:** Statistical results for initial heading directions of experienced (E) and naive (N) dumpers in 2 m north, 2 m east, 2 m south and 2 m west displacement tests. The last line concerns naive dumpers at the test site nearest to where they dumped their waste.

Test	Mean Vector	95% Confidence Interval		Rayleigh Test		V Test Detection 0°	
	µ	Minus	Plus	Z	*p*	Z	*p*
2 m North (E)	333.71°	321.26°	346.16°	12.91	<0.0007	4.557	<0.0002
2 m North (N)	227.94°	179.31°	276.53°	2.81	0.057	−1.59	0.944
2 m East (E)	345.10°	321.82°	8.388°	8.8	<0.0002	4.055	<0.0005
2 m East (N)	328.47°	164.2°	295.88°	0.51	0.61	0.864	0.196
2 m South (E)	358.27°	337.28°	19.27°	9.72	<0.0005	4.409	<0.0004
2 m South (N)	25.69°	312.211°	99.17°	1.55	0.214	1.589	0.056
2 m West (E)	359.92°	341.572°	18.29°	10.82	<0.0001	4.652	<0.0001
2 m West (N)	112.04°	31.736°	192.35°	1.38	0.254	−0.625	0.732
Dumpers at nearest test site (N)	28.69°	314.211°	89.17°	4.82	0.0214	2.89	<0.005

## Data Availability

Supplementary videos, Excel file of data and R scripts are available at Open Science framework: https://osf.io/cwm46//files/ (accessed on 20 January 2024).

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
