# Peer review of "Desert Ant (Melophorus bagoti) Dumpers Learn from Experience to Improve Waste Disposal and Show Spatial Fidelity"

_insects, 2024, doi:10.3390/insects15100814_

Round 1

Reviewer 1 Report

Comments and Suggestions for Authors

This is a carefully executed paper on the behaviour of waste removing ants and how the ants find their the way back to their nest after a wate removal trip. It deserves publication after some revision.

Most of my few comments are minor

A major theme of the paper is the lack of learning walks by the waste removing ants. That suggests that naïve ants learn how to get back to their nest during their early waste carrying excursions and that they look back at the nest during these excursions. The paper would be much improved by seeing whether the supposition of nest fixation on outward trips is correct. The lack of this information is also mentioned in the subsequent line by line comments.

Para beginning line 241. In your data set is there a high correlation between your two measures of straightness and, if so, why do you need 2 and, if not, how does using 2 measures help?

Line 260 scanning bout. Are there fixations in any preferred directions, e.g. the nest? Worth checking.   

Line 331 typo did not engage IN a learning walk.

Line 342 Could you remind readers if that’s correct that no learning walks seen in the period of testing, but walks could have occurred earlier.

Line 439-440. Did you follow whether the naïve ants on taking a long time on their 1st trip did or didn’t decide to take a learning trip on their 2nd or 3rd trip? Line 570 or the 439-440 could make that clear.

Line 585 ‘Learning on the job’ It would be interesting to see the extent to which ants do fixate the nest during their early trips.

Author Response

Reviewer 1:

Comments and Suggestions for Authors

This is a carefully executed paper on the behaviour of waste removing ants and how the ants find their the way back to their nest after a wate removal trip. It deserves publication after some revision.

Response: Thanks for the encouragement.

Most of my few comments are minor

A major theme of the paper is the lack of learning walks by the waste removing ants. That suggests that naïve ants learn how to get back to their nest during their early waste carrying excursions and that they look back at the nest during these excursions. The paper would be much improved by seeing whether the supposition of nest fixation on outward trips is correct. The lack of this information is also mentioned in the subsequent line by line comments.

Response:  This theme is a potential future project that requires both much more effort and a higher frame rate for best determination of looking directions. We have other data of walks or parts of walks at a higher frame rate, but those will need to go into a separate paper. At this point we focusing preliminary differences between the path characteristics of experienced and naïve dumpers.

Para beginning line 241. In your data set is there a high correlation between your two measures of straightness and, if so, why do you need 2 and, if not, how does using 2 measures help?

Response: There is usually a (negative) correlation between straightness and sinuosity, but these two variables measure two different aspects path characteristics, and sometimes we find differences between groups in one measure but not the other. Thus, if a traveller makes big but ‘unwiggly’ loops to the left and right, straightness will be low because the straight-line distance will be small for the path length, but sinuosity will also be low because moment-to-moment wiggles are few and the travel paths are mostly straight.

Line 260 scanning bout. Are there fixations in any preferred directions, e.g. the nest? Worth checking.   

Response: As discussed earlier, we are analysing aspects of scanning, including the duration of scans, in a separate data set with a higher frame rate.

Line 331 typo did not engage IN a learning walk.

Response: Thank you for that, we fixed the typo.

Line 342 Could you remind readers if that’s correct that no learning walks seen in the period of testing, but walks could have occurred earlier.

Response: We do not think it likely that any of the naïve ants had taken any learning walks before taking part in the experiment; that is, we think that they were truly naïve. In the procedure, every ant appearing outside was painted with a dot for 9 days. These ants were not used in the study. If an unpainted ant appearing after 9 days was not naïve, it would have had to come outside and do some work and then stay home for at least 9 days (as the experiment progressed, the number of such days increased). We have not observed any ants doing that in the study or in many previous studies. We now argue this point along these lines in the Methods section. 

Line 439-440. Did you follow whether the naïve ants on taking a long time on their 1st trip did or didn’t decide to take a learning trip on their 2nd or 3rd trip? Line 570 or the 439-440 could make that clear.

Response: We have strictly focused on the ants which not perform any learning walks between the consecutive learning walks. If any ant perform relearning walk we have excluded from the study. Now we have included this in the manuscript (Line 440-442).

Line 585 ‘Learning on the job’ It would be interesting to see the extent to which ants do fixate the nest during their early trips.

Response: As indicated, we are examining these aspects in a separate data set with a higher video frame rate.

Reviewer 2 Report

Comments and Suggestions for Authors

The manuscript presents three interesting experiments that explore the navigational behaviour of individuals that remove waste from the nest in the ant Melophorus bagoti. Overall, the paper is very well written, data visualised very nicely. The findings are novel. Just a few points which I hope the authors can address easily.

The results suggests that learning is involved in the observed differences between ants that were active in the days prior to the experiment, and those that were not. The latter are termed 'naive', however, the rationale for doing could be better justified. It implies that they did not have any experience with navigation outside the nest but the lack of learning walks suggests the contrary. The reader should be reminded of that in the Discussion, when learning is discussed as the most parsimoneous explanation. Prior experience cannot be fully ruled out, and reorientation and task switching could result in similar behavioural outcomes.

Further comments:

Introduction, Methods (Lines 164-178) - the term 'naive' should be defined more clearly. Since none of the naive individuals included in the analysis for the present study performed a learning walk, there seem to be sufficiently familiar with the visual surroundings. However, they were also not seen to exit the nest on a few days. What exactly is the 'naive' state of these ants? Do dumpers ever forage, and if so when and how? Please expand on these questions. You mention a bit in Results (lines 331-342), it would be better to have this in Methods.

Lines 174-175: As written it is unclear whether ants were tested twice, for example initially as naive and later on as experienced foragers. If not, please explain how you selected experienced foragers.

L.145 - State more clearly that each experiment was conducted with one colony.

Methods: Please state which paint was used to mark the ants, and its properties that make it safe to use and durable.

Section 2.3.3 - Please explain the marking procedure in more detail. Did you paint single dots? How distinguishable were the colours? How many individuals were marked, and did each marked ant emerge five times (over which period of time)?

L.193 - Where was the tripod located? If not centred on the nest, how were tracking distances and body orientations over the whole FoV calibrated for the extraction of trajectories? Please add.

L.219-220: Add citation and source link for DLTdv8

Methods and Results (lines 343-344: Expand on the rationale to test only zero-vector ants and rationale and predictions for test releases close to the nest. Most readers may find it difficult to understand these nuances in the design of the experiments.

Methods and Table 1 - add sample sizes for each test and explain whether ants were tested only once or repeatedly. If the latter explain how you estimated the number of individuals tested.

Results Exp 2 and 3 - state sample sizes, and remind the reader where repeated measurements were taken, or whether the same ants could have been tested repeatedly.

Author Response

Reviewer 2: Comments and Suggestions for Authors

The manuscript presents three interesting experiments that explore the navigational behaviour of individuals that remove waste from the nest in the ant Melophorus bagoti. Overall, the paper is very well written, data visualised very nicely. The findings are novel. Just a few points which I hope the authors can address easily.

The results suggests that learning is involved in the observed differences between ants that were active in the days prior to the experiment, and those that were not. The latter are termed 'naive', however, the rationale for doing could be better justified. It implies that they did not have any experience with navigation outside the nest but the lack of learning walks suggests the contrary. The reader should be reminded of that in the Discussion, when learning is discussed as the most parsimoneous explanation. Prior experience cannot be fully ruled out, and reorientation and task switching could result in similar behavioural outcomes.

Response: As argued above, with 9 days of painting any and every ant that emerged outside, an unpainted ant that appeared afterwards is almost certainly naïve. We argue this case now in the Methods.

Further comments:

Introduction, Methods (Lines 164-178) - the term 'naive' should be defined more clearly. Since none of the naive individuals included in the analysis for the present study performed a learning walk, there seem to be sufficiently familiar with the visual surroundings. However, they were also not seen to exit the nest on a few days. What exactly is the 'naive' state of these ants? Do dumpers ever forage, and if so when and how? Please expand on these questions. You mention a bit in Results (lines 331-342), it would be better to have this in Methods.

Response: We now argue for the naïve status of the emerging ants after 9 days of painting.

Lines 174-175: As written it is unclear whether ants were tested twice, for example initially as naive and later on as experienced foragers. If not, please explain how you selected experienced foragers.

Response: Now we have clarified, any of the naive ants’ dumping activities were monitored until they performed over four days, making at least 5 dumping runs. After four days of experience outside, they were considered experienced dumpers. Also, we have mentioned that Each individual experienced or naive dumper was tested only once. To avoid repeated testing, ants were marked with an extra dot of paint on their body.

L.145 - State more clearly that each experiment was conducted with one colony.

Response: Now we have stated that each experiment was conducted at different nest site (Line 147).

Methods: Please state which paint was used to mark the ants, and its properties that make it safe to use and durable.

Response:  Now details were provided in the manuscript (Line 168-169).

Section 2.3.3 - Please explain the marking procedure in more detail. Did you paint single dots? How distinguishable were the colours? How many individuals were marked, and did each marked ant emerge five times (over which period of time)?

Response: More details has been provided in the section (Line 212-218).

L.193 - Where was the tripod located? If not centred on the nest, how were tracking distances and body orientations over the whole FoV calibrated for the extraction of trajectories? Please add.

Response: Tripod is centred at the nest. We have mentioned at Line 199., however we clarified again in the section 2.3.3 again.

L.219-220: Add citation and source link for DLTdv8

Response: (Hedrick, 2008) citations has been added.

Methods and Results (lines 343-344: Expand on the rationale to test only zero-vector ants and rationale and predictions for test releases close to the nest. Most readers may find it difficult to understand these nuances in the design of the experiments.

Response: We chose to test only zero-vector ants to ensure that the ants had no prior outbound experience that could influence their navigational behavior. Zero-vector ants are those that have not yet established a habitual route or developed significant navigational knowledge from previous foraging trips. By focusing on these ants, we aimed to observe the pure learning process and initial navigational strategies employed by naive dumpers. We have provided more information in the manuscript (352-359).

Methods and Table 1 - add sample sizes for each test and explain whether ants were tested only once or repeatedly. If the latter explain how you estimated the number of individuals tested.Results Exp 2 and 3 - state sample sizes, and remind the reader where repeated measurements were taken, or whether the same ants could have been tested repeatedly.

Response: We have mentioned very clearly that in the methodology section.
